# Reaction Pathway Analysis of Methane and Propylene Cracking: A Reactive Force Field Simulation Approach

**DOI:** 10.3390/ma18122672

**Published:** 2025-06-06

**Authors:** Wei Yang, Yiqiang Hong, Youpei Du, Zhen Dai, Guangyuan Cui, Geng Chen, Dabo Xing, Yunlong Ma, Lei Liang, Hongyang Cui

**Affiliations:** Beijing System Design Institute of Mechanical-Electrical Engineering, Beijing 100871, China

**Keywords:** molecular dynamics, ReaxFF, chemical vapor deposition (CVD)

## Abstract

This study presents the development and validation of an elementary reaction pathway tracking algorithm based on reactive force field simulations, enabling the dynamic monitoring of cracking products at the 20,000-atom scale, the accurate identification of chain reaction pathways, and the comprehensive tracking of large carbon chain formation. The research demonstrates that the differences between methane and propylene cracking–polymerization reactions primarily stem from disparities in bond dissociation energies, radical stabilities, and molecular topologies, and the operation of molecular dynamics relies on LAMMPS 3 March 2020. The cracking pathway of methane is relatively straightforward, predominantly involving the homolytic cleavage of C–H bonds, followed by radical chain propagation leading to the formation of large carbonaceous species. In contrast, propylene, owing to its unsaturated structure and multiple reactive sites, exhibits more complex reaction networks and a wider diversity of products. Furthermore, the study elucidates the reaction pathways of intermediate species during methane and propylene cracking and investigates the effect of reaction temperature on carbon sheet development. In conclusion, the algorithm established in this work offers a detailed mechanistic insight into the gas-phase cracking of methane and propylene, providing a new theoretical basis for the optimization of gas-phase deposition processes and the rational design of carbon-based materials.

## 1. Introduction

Methane and propylene are widely utilized as carbon precursors in the fabrication of carbon–carbon (C/C) composites due to their favorable decomposition characteristics and availability [1]. These hydrocarbons are extensively employed in chemical vapor deposition (CVD) processes to produce high-performance C/C composites, which are critical in aerospace, automotive, and energy applications. The CVD process [2] involves the thermal decomposition of hydrocarbon gasses, leading to the deposition of solid carbon on substrates, forming the desired composite material.

Understanding the thermal cracking mechanisms of methane and propylene is essential for optimizing CVD processes [3]. The decomposition pathways, intermediate species, and reaction kinetics directly influence the quality, microstructure, and properties of the resulting C/C composites. For instance, the cracking of methane at high temperatures leads to the formation of various hydrocarbons and solid carbon, affecting the deposition rate and the structural integrity of the composite [4].

Recent advancements in computational modeling, particularly reactive molecular dynamics (MDs) simulations using the ReaxFF force field [5], have provided deeper insights into the complex reaction networks involved in hydrocarbon cracking. ReaxFF allows for the simulation of bond-breaking and bond-forming events, enabling the study of chemical reactions at the atomic level over extended timescales. Studies employing ReaxFF MD simulations [6] have elucidated the decomposition mechanisms of methane and propylene, revealing the formation of various intermediate species and the influence of temperature on reaction pathways [7].

Despite these advancements, challenges remain in accurately capturing the dynamic evolution of reaction intermediates and understanding the synergistic effects of entropy and enthalpy on product distribution during hydrocarbon cracking. Addressing these challenges is crucial for the precise control of CVD processes and the development of C/C composites with tailored properties.

In this study, we develop a reaction pathway tracking algorithm based on atom labeling, coupled with a comprehensive method to trace the formation pathways of large carbon structures. By applying this approach to methane and propylene systems, we uncover the dynamic regulation of cracking and polymerization processes across a range of temperatures. Notably, our findings reveal a non-monotonic competition between entropy and enthalpy [8] in determining product distributions, providing new theoretical insights into the synergistic regulation mechanisms that govern carbon deposition. This work lays a scientific foundation for the rational design and process optimization of C/C composites and contributes to the broader understanding of hydrocarbon chemistry under extreme conditions.

## 2. Computational Model and Methodology

As shown in Figure 1a,b, methane and propylene boxes were prepared for this experiment. To ensure observable reactions between molecules within a sufficiently short timeframe, a box density of 0.5 g/cm^3^ was selected based on the literature [9].

The methane simulation box contained 4000 molecules (60 × 60 × 60 Å^3^, density 0.50 g/cm^3^), while the propylene box comprised 2222 molecules (68 × 68 × 68 Å^3^, density 0.50 g/cm^3^). These dimensions [10] ensure the following: (1) compatibility with ReaxFF’s 12 Å cutoff radius; (2) gas-phase densities matching industrial pyrolysis conditions; and (3) <5% deviation in collision frequencies compared to experimental data.

The study opted for relatively large reaction systems, which, despite increasing the computational resources required for the simulation, allowed for the effective sampling of the reactions, leading to more accurate averages while minimizing boundary effects. We constructed methane and propylene reaction systems using the Packmol v20.14.2 [11] software and performed simulations with the ReaXFF force field [12], implemented in LAMMPS 3 March 2020 [13]. The simulations were conducted in the NVT ensemble [14] at a pressure of one atmosphere, with temperatures set at 1500 K, 2000 K, 2500 K, 3000 K, and 3500 K. The acronym NVT refers to the canonical ensemble in statistical thermodynamics and molecular dynamics (MDs) simulations, where the number of particles (N), volume (V), and temperature (T) are held constant. This approach allowed us to systematically investigate the reaction behavior of methane and propylene across a wide range of temperatures under controlled conditions. Both systems were pre-equilibrated in an NPT ensemble at 600 K and 1 bar for 1000 ps to ensure structural stability, followed by reactions at the target temperatures for 1000 ps in an NPT ensemble. In this paper, based on the temperature range and simulation observation time of the system, the activity level of the cracking reaction was determined. The high-temperature stage was defined as 3000–3500 K, which was only used in this paper for observing the cracking reaction mechanism of the system within the temperature range where the reaction was active. All the algorithms presented in the article were implemented through custom-compiled Python 3.9 scripts [15].

## 3. Results and Discussion

### 3.1. Dynamic Monitoring and Statistical Analysis Algorithm for Methane/Propylene Cracking Products

The high-temperature thermal cracking of methane and propylene is a complex reaction involving numerous intermediates and products, where both the types and concentrations of species continuously evolve over reaction time. As shown in Figure 2, the algorithm flowchart and output structure of the program indicate that this function can be achieved by relying on the ReaxFF/Species naming provided by LAMMPS itself. These reaction products directly influence the kinetics of carbon deposition and the structural growth characteristics of the resulting materials. Consequently, understanding the dynamic distribution of products is essential for optimizing deposition conditions and enhancing the quality of carbon–carbon composite materials.

The dynamic monitoring and statistical analysis algorithm developed for methane and propylene cracking products provides an analytical framework for the real-time tracking and statistical evaluation of product distributions throughout the reaction process. Utilizing the built-in ReaxFF/Species functionality within reactive force field molecular dynamics simulations, the method continuously monitors and records product species in molecular dynamics trajectories, thereby offering real-time insights into reaction progress. Subsequently, custom Python scripts are employed to aggregate and analyze the recorded data, visualizing the results through graphical representations.

Meanwhile, bond dissociation energies [16], radical stabilities, and molecular topologies were used to characterize the cracking mechanisms of methane and propylene, among which, BDEs serve as direct thermodynamic metrics, radical stability reflects electronic delocalization and substituent effects, and molecular topology dictates steric and electronic environments.

By systematically examining changes in product concentrations over time, this method clearly elucidates the generation and consumption patterns of various products during thermal cracking. Moreover, it facilitates the identification of key reaction pathways and dominant species under specific temperature and pressure conditions. Thus, this approach effectively captures the chain reaction behavior underlying the conversion of methane and propylene into multi-carbon compounds, providing valuable data to optimize reaction conditions, precisely control product distributions, and regulate deposition processes. Ultimately, the algorithm provides a foundational methodology for enhancing the microstructural uniformity, density, and mechanical properties of carbon–carbon composite materials.

Based on molecular dynamics simulations and thermodynamic analyses (Figure 3), the differences in the cracking–polymerization behavior of methane (CH_4−_) and propylene (C_3_H_6_) are primarily due to the combined effects of their chemical bond dissociation energies (BDE) [17] radical stability, and molecular topologies. Specifically, the dissociation energies vary significantly across propylene’s C–C single bond (BDE ≈ 347 kJ/mol) [18], C–H bond (BDE ≈ 436 kJ/mol) [18], and C=C double bond (BDE ≈ 615 kJ/mol) [19]. At elevated temperatures (>3000 K), the C–C bond preferentially dissociates, generating smaller radicals such as CH_3_·and C_2_H_3_·. As the temperature continues to rise, the likelihood of the C–H bond dissociation increases, leading to the formation of allyl radicals (C_3_H_5_·), which are stabilized through conjugation, alongside atomic hydrogen (H·). These radicals then facilitate the production of larger molecules (C_4−_ and above) via chain propagation (e.g., C_3_H_5_· + C_3_H_6_ → C_6_H_11_·) or bimolecular coupling reactions (e.g., 2C_3_H_5_· → C_6_H_10_), this work highlights the critical role of allyl radical stabilization through conjugated systems in propylene cracking, as evidenced by reduced methane formation and increased C_3_H_5_· intermediate lifetimes. In contrast, the carbon atoms of methane adopt SP3 hybridization, forming a completely symmetrical tetrahedral structure, which evenly distributes the electron clouds of the four C–H bonds and avoids local planning. The elevated C–H bond dissociation energy (BDE) of methane (460 ± 15 kJ/mol for CH_4_ → CH_3_ + H·) compared to propylene (e.g., allylic C–H BDE ≈ 364 kJ/mol) indicates that methane has a higher C–H bond energy. Compared to the allylic electronic conjugation stability structure conferred by the double bond in propylene, methane indeed has additional stability due to its structure, the kinetic modeling of methane flames reveals that C_2_–C_4−_ intermediates dominate methane pyrolysis due to its high C–H BDE (~439 kJ/mol), leading to limited radical lifetimes and rapid recombination (CH_3_·→ CH_4−_) [20]. As a result, its radicals (CH_3_, CH_2_·) have shorter lifetimes and higher recombination rates, which leads to lower cracking efficiency and a narrower range of product distributions compared to propylene. Experiments in a jet-stirred reactor show that methane suppresses n-heptane low-temperature oxidation by scavenging OH and HO_2_ radicals. This study quantifies methane’s role in radical recombination (e.g., CH_3_· + OH → CH_4−_ + O·), which shortens radical chain propagation [21].

Temperature has a significant impact on bond dissociation rates and free radical concentrations during the cracking–polymerization process. At relatively low temperatures (<2500 K), propylene primarily yields C_3_ species, indicating limited molecular fragmentation and a low overall cracking rate. At intermediate temperatures (~3000 K), the increased dissociation of C–C bonds generates numerous C_1_ and C_2_ radicals. These highly reactive intermediates accelerate global chain reactions, promoting the formation of larger fragments (up to C_10_ and beyond). At higher temperatures (3000–3500 K), an excess of reactive intermediates (C_1_, C_2_ radicals) further extends chain propagation, resulting in the generation of larger molecular fragments. Simultaneously, higher molecular weight radical intermediates combine to form stable molecular species, effectively terminating reaction pathways. These stable molecules contribute to the system’s product diversity and serve as new precursors for subsequent cracking reactions. Shao et al. attribute this to methane’s tetrahedral geometry, which restricts π-conjugation and stabilizes fewer radicals compared to propylene [17].

Furthermore, differences in molecular topology play a crucial role in reaction network complexity. The relationship between the unsaturated structure of propylene and radical resonance stability is fundamentally rooted in the conjugation effects enabled by its sp^2^-hybridized planar geometry. Propylene’s π-electron system allows the delocalization of unpaired electrons in allylic radicals (e.g., C_3_H_5_), stabilizing these intermediates through resonance [18], propylene’s planar, unsaturated structure allows for resonance stabilization of radicals (such as C_3_H_5_·, stabilization energy ≈ 25 kJ/mol), which prolongs radical lifetimes and facilitates their involvement in chain reactions, even at lower temperatures (1500~2500 K). Tian et al. confirm that propylene’s planar topology and π-conjugation enhance radical stability (~25 kJ/mol), enabling sequential oligomerization (e.g., 2C_3_H_5_· → C_6_H_10_), while methane’s rigid geometry prevents analogous mechanisms [19]. In contrast, methane’s tetrahedral structure restricts its reaction network primarily to a simple linear chain mechanism (CH_4−_ ↔ CH_3_· ↔ CH_2_·). Consequently, radical recombination reactions dominate methane cracking, resulting in limited complexity and product diversity. Even at temperatures as high as 3000 K, methane predominantly produces small C_1_ species, limiting potential molecular growth. Jin et al. [20]. employ kinetic modeling to analyze methane combustion, revealing that methane pyrolysis predominantly generates C_2_–C_4−_ intermediates due to its high C–H bond dissociation energy (BDE ≈ 439 kJ/mol). The rapid recombination of methane-derived radicals (e.g., CH_3_· → CH_4−_) suppresses chain propagation, aligning with our observation of methane’s limited product diversity and the dominance of small C_1_ species even at high temperatures (~3000 K). In contrast, resonantly stabilized radicals like propargyl (C_3_H_3_·) in propylene systems exhibit prolonged lifetimes, facilitating polycyclic aromatic hydrocarbon (PAH) formation via bimolecular coupling (e.g., 2C_3_H_3_· → C_6_H_6_).

In conclusion, the differences in cracking–polymerization behavior between methane and propylene are mainly attributed to variations in bond dissociation energies, radical stabilities, and molecular topologies. Propylene’s distinct C–C, C–H, and C=C bond dissociation energies facilitate the formation of reactive radicals at elevated temperatures, which undergo chain propagation and bimolecular coupling, producing higher molecular weight products. Methane, on the other hand, exhibits higher C–H bond dissociation energy and structural constraints, limiting its reaction pathways and leading to lower cracking efficiency and limited product distributions. Temperature significantly affects both systems; as temperature increases, propylene exhibits progressively richer product diversity and higher molecular weight, whereas methane maintains relatively simple reaction characteristics. Additionally, the molecular topology of propylene enhances radical stability via resonance, enabling effective chain reactions even under milder thermal conditions. In contrast, methane’s tetrahedral geometry limit reaction diversity, resulting in fewer and simpler products.

### 3.2. Methane/Propylene Chain Reaction Pathway Identification and Statistical Analysis Algorithm

The primary objective of the atomic labeling algorithm is to accurately trace complex chemical reaction networks at the atomic scale, providing an efficient approach for identifying reaction pathways (Figure 4). This is achieved by assigning unique identifiers to individual atoms within the reaction system, which allows for the detailed tracking of both unimolecular and bimolecular reactions throughout the entire process.

In the Lammp_1 module, methane pyrolysis reactions are simulated using the LAMMPS 3 March 2020 [13] producing kinetic trajectories of the system under high-temperature conditions. Coordinate information is recorded over the entire reaction timescale by configuring the Dump parameter, resulting in trajectory outputs in the XYZ file format.

In the Python_1 module, the atomic labeling algorithm assigns and maintains unique identifiers for each atom, establishing correspondences between atoms and molecules before and after reactions. The dictionaries Atom_Mothermol_Dic and Mothermol_Atom_Dic are utilized to accurately map relationships between reactant molecules and their constituent atoms. After product formation, the dictionary Mol_Atom_Dic is created to map product molecules to their corresponding atoms. Subsequently, the algorithm generates the Mol_Mother_Dic dictionary based on atomic identifiers, explicitly defining the reactant–product relationships. These relationships form the foundational basis for constructing and balancing chemical equations for reaction chains.

In the Python_2 module, the Single and Bimolecular Recognition Algorithm is implemented through a series of interconnected dictionaries (Figure 5). Initially, the Mothermol_Atom_Dic dictionary is extracted from the n.xyz file, followed by the construction of the Mol_Atom_Dic dictionary from the (n+1).xyz file. By comparing these two dictionaries, the Sonmol_Atom_Dic dictionary is generated. This process involves randomly selecting a Mol molecule and iterating over its constituent atoms. Atoms originating from the same Mothermol are grouped together and collectively defined as Sonmol. Thus, a Sonmol consists of atomic fragments derived from the same original Mothermol molecule. Using the atomic labeling algorithm, the relationships between Mol, Sonmol, and Mothermol are precisely established.

As shown in Figure 5, the design purpose of the Mother_Atom_Dic is to statistically record the composition information of molecules and atoms in the nth frame. In this frame, the molecules serving as reactants are counted. The design purpose of the Mol_Atom_Dic is to statistically record the molecular and composition information of the products in the nth + 1 frame. In this frame, the molecules serving as products are counted. Fragments from the same reactant in the products are defined as Son_Atom_Dic, meaning the fragments from the same parent in the products. I think the illustration is very clear in explaining this point. The comparison dictionary used is the Mother_Atom_Dic and the Mol_Atom_Dic. By comparing the atomic composition information, the distribution of reactant molecular fragments in the products is identified, and through algorithmic statistics, the Son_Atom_Dic is formed.

The construction and balancing of chemical equations are based on the relationships identified among Mol, Sonmol, and Mothermol. First, the molecular formula of Mol is placed on the right side of the chemical equation. Then, based on the established Mol–Sonmol–Mothermol relationships, the corresponding Mothermol is placed on the left side, forming an initial unbalanced chemical equation. The next step involves balancing the equation according to the principle of atomic conservation. The number of carbon atoms on both sides of the equation is quantified (denoted by C_r_ and C_l_ for the right and left sides, respectively). If C_r_ > C_l_, additional carbon atoms (C) are added to the left side to balance the difference, specifically introducing C_(r–l)_ atoms. Hydrogen atoms (H) are balanced in a similar manner. If the number of carbon atoms is already balanced, the hydrogen atoms will determine the direction of balancing.

As shown in Figure 6, the reaction networks of methane (CH_4−_) and propylene (C_3_H_6_) in the gas phase both follow a free-radical chain mechanism. However, the differences in their pathway branching and product distribution fundamentally stem from differences in molecular topology and bond energies. From the perspective of molecular topology and bond energies, the saturated sp^3^ hybridization of methane and its tetrahedral symmetry limit the number of reactive sites. The saturated sp^3^ hybridization of methane and its tetrahedral symmetry impose intrinsic constraints on reactivity by enforcing uniform electron distribution across four equivalent C–H σ-bonds (bond length: 0.110 nm, bond angle: 109.5°). This structural rigidity eliminates preferential reactive sites and elevates C–H bond dissociation energy (BDE) to ~460 kJ/mol, requiring extreme thermal conditions (>2500 K) for homolytic cleavage. ReaxFF simulations demonstrate that methane-derived methyl radicals (CH_3_·) exhibit short lifetimes (<1 ps at 1500 K) due to rapid recombination, while propylene’s allylic radicals persist via resonance stabilization [22]. The stability of its free radicals (such as CH_3_· and CH_2_·) is low, and these radicals can only undergo dehydrogenation or recombination reactions through a linear chain cycle initiated by H atoms (e.g., H + CH_4−_ → CH_3_ + H_2_). Consequently, the products primarily consist of CH_4−_ and its derivatives (such as CH_3_ and CH_2_), with a narrow distribution of molecular weights.

In contrast, the molecular structure of propylene, with its unsaturated sp^2^ hybridization (C=C double bond), provides two reactive sites. Propylene’s sp^2^-hybridized C=C double bond provides two reactive sites—allylic C–H bonds and the π-system—due to its electronic delocalization, planar geometry, and reduced bond dissociation energy. These sites are experimentally validated in catalytic oxidation [23], polymerization [24], and pyrolysis kinetics, as reported in the study, methane’s saturated sp^3^ structure lacks such spatially and electronically distinct regions, underscoring propylene’s unique reactivity in chain reactions [25].

The weak polarization of the C=C bond [17,26] facilitates the attack of H atoms, generating allyl radicals (C_3_H_5_·). The delocalization of π electrons, through σ-π hyperconjugation, stabilizes intermediates (such as C_3_H_3_·), significantly extending the lifetime of chain propagation and supporting competitive reactions such as cracking, polymerization, and cyclization, Zhang et al. [26] highlight how zeolite topology enhances allyl radical stabilization via confinement effects, enabling C_3_H_5_· intermediates to participate in chain propagation (C_3_H_5_· + C_3_H_6_ → C_6_H_11_).

The further analysis of the temperature dependence of the chain initiation reactions for methane and propylene yields the following results:

#### 3.2.1. Temperature Dependence of Chain Initiation Reactions in the Methane System

Low Temperature (1500–2000 K): The reaction is dominated by recombination, with the primary reactions being the recombination of H atoms (2H → H_2_) and minor methane dissociation (CH_4−_ → H + CH_3_). The free radical concentration is low, and the recombination rate exceeds the dissociation rate.

Medium to High Temperature (2500–3000 K): Dissociation reactions increase significantly, with the key reaction CH_4−_ → H + CH_3_ accelerating, accompanied by the formation of C_2_H_6_ (2CH_3_ → C_2_H_6_). While the free radical concentration increases, recombination reactions (H + CH_3_ → CH_4−_) still limit the progress of polymerization reactions.

High Temperature (3000–3500 K): Cracking and polymerization reactions reach equilibrium, producing a large number of C_1_, C_2_, and C_3_ free radicals, along with other intermediates involved in chain reactions. While the formation of higher molecular weight products increases, methane cracking still follows a single reaction pathway.

#### 3.2.2. Temperature Dependence of Chain Initiation Reactions in the Propylene System

Low Temperature (1500–2000 K): Chain initiation reactions begin, with the primary reactions being the hydrogenation/dehydrogenation equilibrium of C_3_H_6_. The C=C double bond remains intact, and the system primarily consists of propylene derivatives (such as C_3_H_4−_ and C_3_H_5_·), with propylene remaining highly stable at low temperatures.

Medium Temperature (2500–3000 K): Cracking reactions start, producing a large number of C_1_, C_2_, and C_3_ free radicals, and the formation of C_4−_ and larger products increases sharply. At this stage, chain reactions within the system extend, and the temperature window for polymerization reactions gradually opens.

High Temperature: Cracking and polymerization reactions compete. Extreme cracking reactions, such as C_2_H_1_ → H + C_2_, generate ultra-small carbon clusters. At this stage, the rate of C–C bond dissociation and the rate of product polymerization gradually compete, suppressing polymerization growth.

The thermal cracking reactions of methane and propylene are significantly influenced by differences in molecular structure and bond energies. Methane’s reaction primarily relies on the homolytic cleavage of C–H bonds and generates products such as ethane through free radical chain growth, with relatively simple reaction pathways and a concentrated product distribution. Varghese et al. [27] confirm that methane’s sp^3^ hybridization and tetrahedral symmetry limit reactive pathways to linear chain mechanisms (e.g., CH_4−_ → CH_3_· + H·). In contrast, propylene, due to its unsaturated structure and dual reactive sites, exhibits more chain initiation pathways, involving both C–C bond cleavage and C–H bond homolysis, resulting in more complex reaction pathways and a broader range of products, Li et al. [28] confirm that higher temperatures (>3000 K equivalence) shift the balance toward extreme cracking (e.g., C_2_H_3_· → C_2_ + H·). These differences directly determine the reaction characteristics of methane and propylene under varying temperature conditions, particularly the competitive relationship between cracking and polymerization reactions, further revealing the dynamic reaction mechanisms of the two systems under high-temperature environments.

### 3.3. Comprehensive Reaction Pathway Tracking Algorithm for Intermediates in Methane Cracking

During the methane pyrolysis process, the generation and evolution of intermediate products directly influence the distribution of final products and the microstructural characteristics of carbon material deposition (Figure 7). Given that the pyrolysis reaction involves multiple pathways and a variety of products, tracking the pathways of intermediate products can reveal the dominant reaction pathways, the lifecycle of key intermediates, and their behavior under different conditions. This algorithm enables the identification of the contribution of intermediate products throughout the entire reaction process and allows for the quantitative analysis of their conversion rates and disappearance mechanisms. Especially in the study of high-temperature (3000~3500 K) pyrolysis reactions, this algorithm provides detailed data on the generation efficiency and pathway selectivity of intermediate products, thereby supporting the optimization of carbon deposition processes and improving the uniformity and density of the materials.

In the Lammp_1 module, the methane pyrolysis reaction is solved using the LAMMPS/ReaxFF module to obtain the kinetic trajectory at high temperatures. The Dump parameter is used to output the coordinate information for the entire reaction time series, generating an XYZ format file. In the Python_1 module, all trajectory files are processed, and molecules are constructed based on bonding rules to establish the Mol_Atom_Dic dictionary. Upon entering the first frame, all atoms are named, and the Atom_Name_Set dictionary is created, which cannot be modified. Upon reaching the final frame, molecules are reconstructed according to bonding rules, and the Mol_Name_Dic is created using Atom_Name_Set. The mass of all molecules is then calculated, and the Weight_Mol_Dic is formed, followed by entry into the fragment pathway analysis module.

In the Python_2 module, the Weight_Mol_Dic is traversed to select the intermediates with the highest molecular weights. In this study, the top three largest molecules are chosen, and the Mmol_Matom_Dic is constructed. Using the atomic naming standard from the first frame (Atom_Name_Set) and the intermediates from the final frame (Mmol_Matom_Dic), all trajectory files are processed. In the selected n.xyz file, molecules are constructed based on bonding rules and named using Atom_Name_Set. Additionally, the Matom_Mothermol_Dic and Matom_Sonmol_Dic for the current structure are constructed using Mmol_Matom_Dic. Here, Mothermol represents the molecular structure in the current frame where the Matom resides; this molecule may consist of the target Matom and other molecules. Since Mmol is generated by multiple Matom atoms, and these Matoms may belong to different Mothermol structures across frames, the fragment formed by Matom atoms of the same Mmol is referred to as Sonmol.

The evolution of large molecular weight fragments in the methane and propylene systems reveals the dynamic balance mechanism between cracking and polymerization, as well as its temperature dependence, particularly the coupling of thermodynamic and kinetic factors (Figure 8). Figure 8 shows that the molecular weight distribution of large-sized fragments in both the methane and propylene systems is primarily governed by the competition between cracking and polymerization mechanisms, regulated by Gibbs free energy (ΔG = ΔH − TΔS). The evolution process exhibits a pronounced three-stage temperature dependence.

#### 3.3.1. Low-Temperature Range (1500–2000 K)

At low temperatures, ΔG > 0 (endergonic), favoring polymerization [29] and the cracking degree of methane and propylene is relatively low, and the reaction is in the kinetic control zone, characterized by a lower bond dissociation rate. The system primarily undergoes recombination reactions. Methane follows a single cracking pathway, with high-activity recombination reactions dominating. In contrast, propylene undergoes homolytic C–H bond dissociation to form unsaturated C=C bonds, and the delocalization of π electrons stabilizes the allyl radical (C_3_H_5_·), extending its lifetime. Additionally, the β-homolytic dissociation of the C-C single bond in propylene generates vinyl and methyl groups, further promoting the formation of large carbon fragments. As a result, propylene can generate C_6_ structures earlier than methane at low temperatures. During this stage, both thermodynamic and kinetic factors are unfavorable for the reaction to proceed.

#### 3.3.2. Medium-Temperature Range (2500–3000 K)

As the temperature increases, ΔG ≈ 0 (equilibrium), resulting in competitive cracking and recombination. Methane undergoes cracking to generate C_1_ radicals, while polymerization requires monomer collisions, which are influenced by the system’s entropy effects. Although the reaction is kinetically favorable, the growth of large fragments remains limited. In contrast, in the propylene system, large molecular carbon fragments can reach sizes of C_10_ and beyond, indicating that temperature increases the concentration of free radicals, thereby reaching the threshold necessary to trigger global chain reactions. Large molecular weight fragments rapidly converge in the later stages of the reaction, further suggesting that the cracking and polymerization rates approach dynamic equilibrium. This temperature range is identified as the optimal temperature window.

#### 3.3.3. High-Temperature Range (≥3500 K)

At high temperatures, ΔG < 0 (exergonic cracking), resulting in dominant C–C bond fission. Both the methane and propylene systems exhibit similar trends in the formation of large molecular carbon fragments. Although the generation rate of large molecular fragments further increases at 3500 K, fluctuations in molecular weight during the later stages indicate that cracking reactions inhibit the ordered polymerization of large carbon fragments. The competition between cracking and polymerization reactions results in the highest molecular weight being lower than that observed at 3000 K, suggesting that at high temperatures, kinetics are no longer the limiting factor in the reaction’s progress. However, high temperatures significantly enhance the system’s entropy effects, which suppresses the ordered polymerization of large carbon fragments. As a result, the reaction equilibrium at 3500 K is predominantly influenced by the entropy effect.

The temperature dependence of methane and propylene cracking–polymerization reactions illustrates how the competition between cracking and polymerization at different temperature intervals influences the formation of large molecular weight carbon fragments. At low temperatures, the reaction is constrained by both thermodynamic and kinetic factors. In the medium-temperature range, propylene systems exhibit favorable temperature window effects for chain reactions and the formation of large molecular weight fragments. At high temperatures, although the reaction is no longer kinetically limited, entropy effects dominate the reaction equilibrium, restricting the ordered polymerization of large carbon fragments. These findings are crucial for optimizing the cracking–polymerization process and for further guiding the microscopic structural control of carbon material deposition.

## 4. Conclusions

In this study, the dynamic behavior of methane and propylene cracking–polymerization reactions was systematically analyzed using advanced molecular dynamics simulations and a series of custom algorithms for the real-time monitoring and statistical analysis of product distributions. The results highlight the crucial role of temperature in shaping the competition between cracking and polymerization pathways, significantly influencing the formation and distribution of large molecular weight carbon fragments. At low temperatures, the reaction is primarily controlled by recombination reactions, with both thermodynamic and kinetic factors limiting the formation of large fragments. As the temperature increases, the propylene system demonstrates a broader range of product formation, driven by the enhanced concentration of free radicals and the initiation of global chain reactions. At high temperatures, although the reaction becomes less kinetically constrained, the system’s entropy effects dominate, limiting the ordered polymerization of large carbon fragments and leading to a dynamic equilibrium between cracking and polymerization processes.

The developed algorithms offer a comprehensive framework for tracking and analyzing the complex reaction networks of methane and propylene, enabling the identification of key intermediates and reaction pathways. These findings are crucial for the optimization of cracking–polymerization processes, providing valuable insights into the regulation of product distributions and the microstructural characteristics of carbon material deposition. By understanding the temperature-dependent behavior and reaction dynamics of these systems, this research lays the foundation for the controlled synthesis of carbon–carbon composite materials with enhanced uniformity, density, and mechanical properties. Ultimately, this work contributes to advancing our knowledge of high-temperature pyrolysis reactions and offers promising strategies for optimizing the production and application of carbon materials.

## Figures and Tables

**Figure 1 materials-18-02672-f001:**
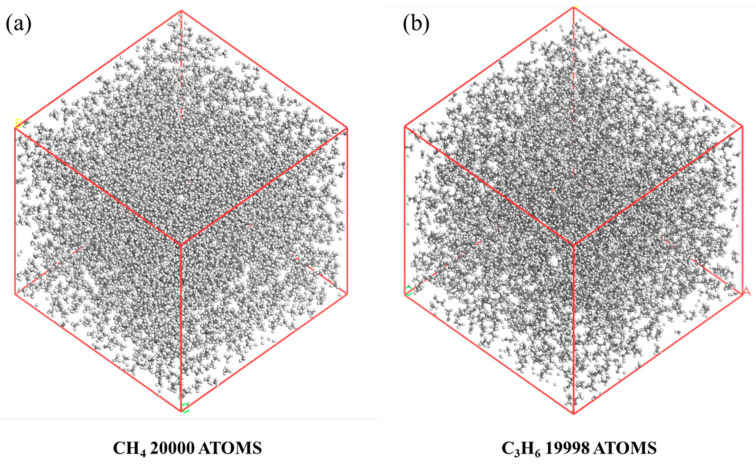
Reaction models: (**a**) methane; (**b**) propene.

**Figure 2 materials-18-02672-f002:**
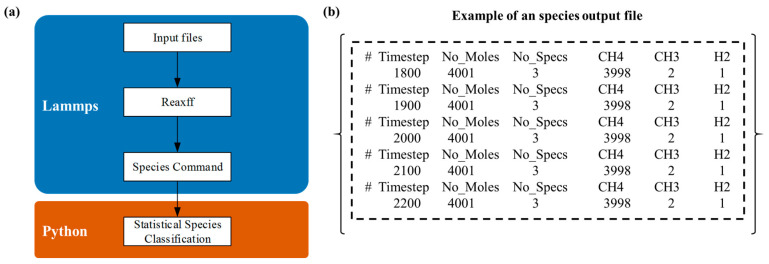
Product statistical analysis algorithm: (**a**) flowchart of the product statistical analysis algorithm; (**b**) example of the species output file.

**Figure 3 materials-18-02672-f003:**
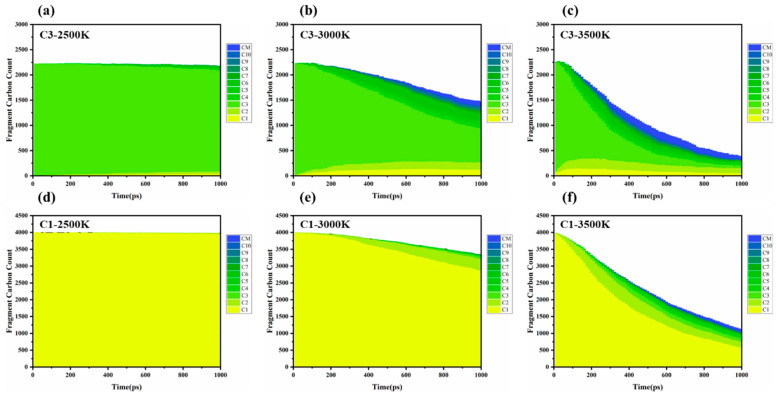
Product monitoring and statistical analysis: Statistics of products of propylene and methane at different temperatures, where CM refers to carbon fragments with more than 10 carbon atoms; (**a**) C3-2500K; (**b**) C3-3000K; (**c**) C3-3500K; (**d**) C1-2500K; (**e**) C1-3000K; (**f**) C1-3500K.

**Figure 4 materials-18-02672-f004:**
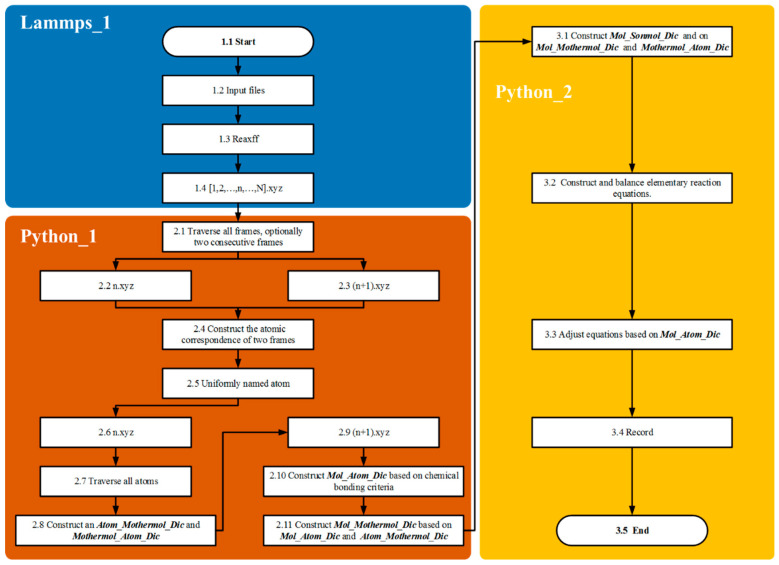
Methane chain reaction pathway identification and statistical analysis algorithm: algorithm flowchart.

**Figure 5 materials-18-02672-f005:**
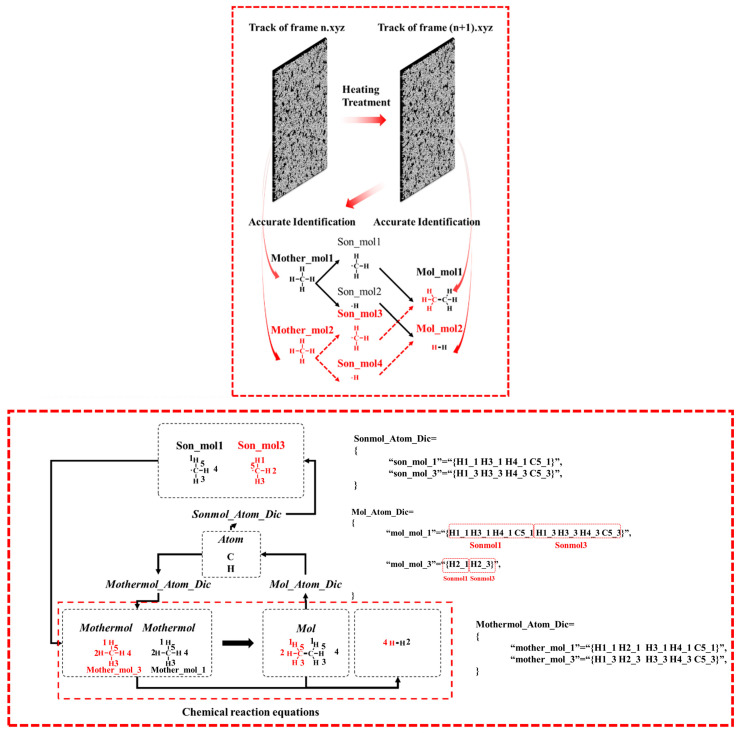
Methane chain reaction pathway identification and statistical analysis algorithm: single/multi-molecule recognition algorithm, the same character color indicates that the molecular fragments come from the same molecule, the above figure is the flowchart of the algorithm, and the below figure is the mechanism diagram of the algorithm.

**Figure 6 materials-18-02672-f006:**
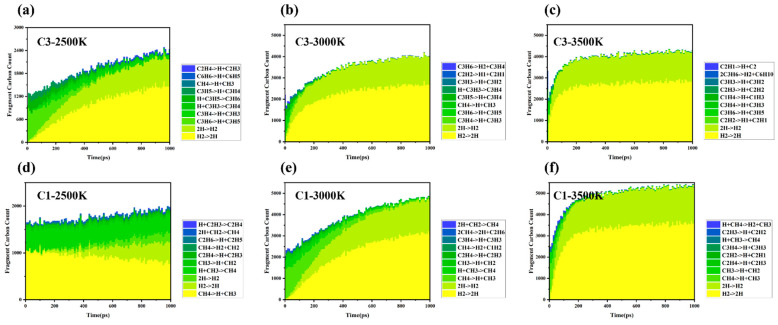
Comprehensive reaction pathway identification and statistical analysis of all methane species; (**a**) C3-2500K; (**b**) C3-3000K; (**c**) C3-3500K; (**d**) C1-2500K; (**e**) C1-3000K; (**f**) C1-3500K.

**Figure 7 materials-18-02672-f007:**
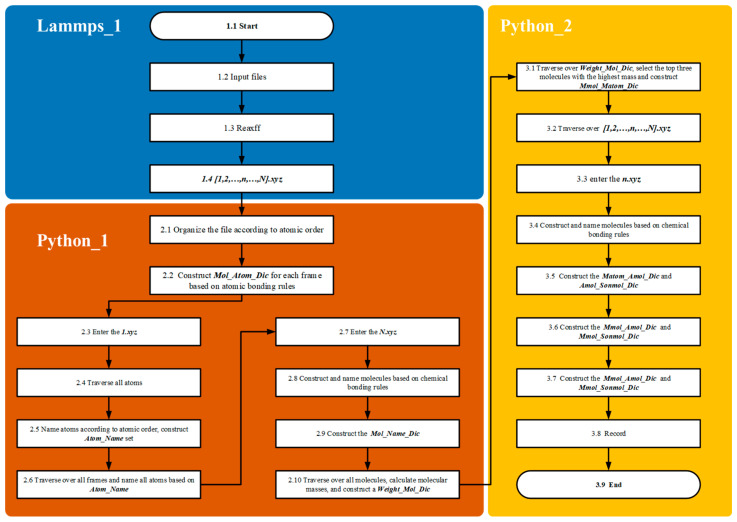
Comprehensive reaction pathway tracking algorithm for: intermediates in methane chain reactions: algorithm flowchart.

**Figure 8 materials-18-02672-f008:**
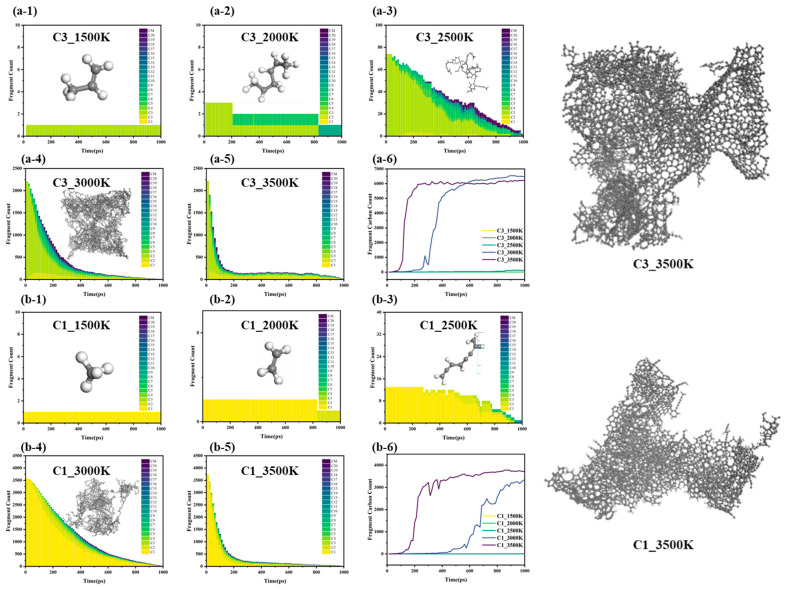
Comprehensive reaction pathway tracking algorithm for intermediates in methane chain reactions; (**a-1**) C3-1500K; (**a-2**) C3-2000K; (**a-3**) C3-2500K; (**a-4**) C3-3000K; (**a-5**) C3-3500K; (**a-6**) Summary of C3; (**b-1**) C1-1500K; (**b-2**) C1-2000K; (**b-3**) C1-2500K; (**b-4**) C1-3000K; (**b-5**) C1-3500K; (**b-6**) Summary of C1.

## Data Availability

The original contributions presented in this study are included in the article/Appendix A. Further inquiries can be directed to the corresponding authors.

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
