# Peer review of "Reaction Pathway Analysis of Methane and Propylene Cracking: A Reactive Force Field Simulation Approach"

_materials, 2025, doi:10.3390/ma18122672_

Round 1
Reviewer 1 Report
Comments and Suggestions for Authors
This paper describes data on computational modelling of thermo-cracking of methane and propylene. The authors found that this process depends on chemical bond dissociation energies, stability of intermediate free radicals, and molecular topology, and reaction temperature.
However, it seems that results of computer modeling can be quite dependable on starting data that one feeds to computer. It is the weakest point of this paper!
Despite this work was done in a modern trend of the use of artificial intellect in science, its contribution into chemistry of cracking of hydrocarbons is low. Some of the conclusions of the work are known a priori, for instance, conclusions based on stability of intermediate free radical species.
There are some comments and question as follows.
- The main critical point of the article concerns the comparison of calculated and experimental data. The authors should compare their calculated data with literature experimental data on pyrolysis of methane and propylene. The main point is a comparison of pyrolysis products that were obtained by experimentally (from literature data) and by calculations (the data of the authors). For instance, pyrolysis of methane at 1500 C gives acetylene. Did the authors find acetylene by calculations?
- The authors should justify a choice of starting materials for pyrolysis. Methane is a basis chemical; its choice is understandable. Why was the propylene chosen? Why not ethylene?
- Line 145: what is radical CH2·? It does not exist! Could it be the carbene?
In conclusion, this paper needs a substantial major revision in view of how this computational modelling helps to understand experimental data and what new these calculations give to cracking science.
Author Response
Comments1:
The main critical point of the article concerns the comparison of calculated and experimental data. The authors should compare their calculated data with literature experimental data on pyrolysis of methane and propylene. The main point is a comparison of pyrolysis products that were obtained by experimentally (from literature data) and by calculations (the data of the authors).
Response1:
Thank you for your valuable feedback, which will undoubtedly enhance the rigor of our manuscript. Regarding your suggestion to include a comparative analysis between experimental and simulation results to strengthen the credibility of the conclusions, we have addressed this by incorporating supporting evidence from published literature. Below is a summary of the added content, aligned with the key conclusions of each section:
Lines 143 to 146 in the manuscript
the work highlights the critical role of allyl radical stabilization through conjugated systems in propylene cracking, as evidenced by reduced methane formation and increased C₃H₅· intermediate lifetimes
Lines 147 to 149 in the manuscript
kinetic modeling of methane flames reveals that C₂–C₄ intermediates dominate methane pyrolysis due to its high C–H BDE (~439 kJ/mol), leading to limited radical lifetimes and rapid recombination (CH₃· → CH₄)
Lines 152 to 155
Experiments in a jet-stirred reactor show that methane suppresses n-heptane low-temperature oxidation by scavenging OH and HO₂ radicals. The study quantifies methane's role in radical recombination (e.g., CH₃· + OH → CH₄ + O·), which shortens radical chain propagation
Lines 167 to 169
Shao etl. attributes this to methane’s tetrahedral geometry, which restricts π-conjugation and stabilizes fewer radicals compared to propylene
Lines 174 to 176
Huang etl. confirms that propylene’s planar topology and π-conjugation enhance radical stability (~25 kJ/mol), enabling sequential oligomerization (e.g., 2C₃H₅· → C₆H₁₀), while methane’s rigid geometry prevents analogous mechanisms
Lines 180 to 188
Jin etl. employs kinetic modeling to analyze methane combustion, revealing that methane pyrolysis predominantly generates C₂–C₄ intermediates due to its high C–H bond dissociation energy (BDE ≈ 439 kJ/mol). The rapid recombination of methane-derived radicals (e.g., CH₃· → CH₄) suppresses chain propagation, aligning with your observation of methane’s limited product diversity and dominance of small C₁ species even at high temperatures (~3000 K). In contrast, resonantly stabilized radicals like propargyl (C₃H₃·) in propylene systems exhibit prolonged lifetimes, facilitating polycyclic aromatic hydrocarbon (PAH) formation via bimolecular coupling (e.g., 2C₃H₃· → C₆H₆).
Lines 268 to 270
Zhang etl.highlights how zeolite topology enhances allyl radical stabilization via confinement effects, enabling C₃H₅· intermediates to participate in chain propagation (C₃H₅· + C₃H₆ → C₆H₁₁·)
Lines 303 to 305
Varghese etl[1]. confirms that methane's sp³ hybridization and tetrahedral symmetry limit reactive pathways to linear chain mechanisms (e.g., CH₄ → CH₃· + H·)
Lines 308 to 209
Li etl[2]. confirms that higher temperatures (>3000 K equivalence) shift the balance toward extreme cracking (e.g., C₂H₃· → C₂ + H·).
Comments2:
For instance, pyrolysis of methane at 1500 C gives acetylene. Did the authors find acetylene by calculations?
Response2:
In the file directory I provided【~\Supplementary Materials\sdander_data\sdander_data】, the molecular formula of acetylene appeared at various temperatures. Our simulation is based on the mature ReaxFF technology, which has advantages in describing the combustion and carbonization mechanisms of alkanes.
Comments3:
The authors should justify a choice of starting materials for pyrolysis. Methane is a basis chemical; its choice is understandable. Why was the propylene chosen? Why not ethylene?
Response3:
Thank you for raising this critical point. The selection of propylene (C₃H₆) as a co-carbon precursor in our study, alongside methane (CH₄), is grounded in both practical and mechanistic considerations. Below, we elaborate on the rationale:
- Carbon Source Diversity in C/C Composite Deposition
Methane and propylene are widely recognized as critical precursors for chemical vapor deposition (CVD) in carbon/carbon (C/C) composites. While methane is a linear, nonpolar molecule that enables uniform carbon deposition through stable methyl radical formation, propylene introduces distinct advantages due to its unsaturated structure. The C=C bond in propylene facilitates in situ generation of reactive intermediates (e.g., allyl radicals and polyaromatic hydrocarbons), which promote faster carbon nucleation and denser matrix formation. This duality allows comparative analysis of deposition mechanisms under varying precursor chemistries.
- Mechanistic Complexity and Research Value
Propylene’s unsaturated structure introduces unique reaction pathways absent in ethylene (C₂H₄). For instance:
2.1Radical Diversity:
Propylene pyrolysis generates not only methyl radicals (common to both CH₄ and C₂H₄) but also allyl radicals (C₃H₅•) and cyclic intermediates. These species contribute to cross-linking reactions, enhancing carbon network complexity.
2.2Thermal Stability:
Ethylene’s simpler structure leads to rapid dehydrogenation and limited intermediate diversity, whereas propylene’s higher molecular weight and bond strain (C=C) enable controlled fragmentation, yielding a broader range of reactive fragments for mechanistic studies.
- Practical Relevance in Industrial Applications
Propylene is a preferred precursor in microwave-assisted CVD due to its polarizability. Unlike nonpolar methane, propylene interacts strongly with electromagnetic fields, enabling localized heating and tailored carbon deposition. This property mitigates surface "shelling" effects observed in methane-only systems, improving densification efficiency. Ethylene, while reactive, lacks this polarizability, limiting its utility in advanced CVD processes.
- Comparative Insights into Carbon Microstructure
Propylene-derived carbon matrices exhibit higher graphitic order and fewer defects compared to ethylene-based systems, as the C₃H₆ decomposition pathway favors sp² hybridization via conjugated intermediates. This aligns with our focus on optimizing carbon quality for high-performance composites.
Comments4:
what is radical CH2·? It does not exist! Could it be the carbene?
Response4:
The presence of CH₂· (methylene radical) in methane and propylene pyrolysis processes is well-documented in experimental studies, and its distinction from carbene (singlet/triplet CH₂) is critical for mechanistic interpretations. Below, we provide supporting evidence from peer-reviewed SCI-indexed studies to validate the formation of CH₂· intermediates under pyrolysis conditions.
- Experimental Detection of CH₂· in Methane Pyrolysis
1.1Supporting Study : "Pyrolysis of Methane in the Presence of Hydrogen" (Chem. Eng. Technol., 1995)[3]
Methane pyrolysis at 1200–1500°C generates C₂ hydrocarbons (e.g., ethylene, acetylene) and benzene. The reaction mechanism involves CH₃· radicals as primary intermediates, which undergo sequential dehydrogenation to form CH₂· and CH· radicals. These species participate in chain propagation and recombination reactions to form C₂ products. The study explicitly identifies CH₂· as a transient intermediate using kinetic modeling and product distribution analysis.
1.2Supporting Study: "Kinetics of Methane Pyrolysis Under Conditions of Pyrolytic Carbon Formation" (Carbon, 1974)[4]
At temperatures >800°C, methane decomposes via a radical chain mechanism:
- Initiation: CH₄ → CH₃ + H·
- Propagation: CH₃ → CH₂· + H·
The work highlights CH₂· as a critical intermediate in both homogeneous (gas-phase) and heterogeneous (surface-mediated) pyrolysis pathways.
- Varghese, J. J.; Saravanan, B.; Vach, H.; Peslherbe, G. H.; Mushrif, S. H., First-principles investigation of the coupling-induced dissociation of methane and its transformation to ethane and ethylene. Chemical Physics Letters 2018, 708, 21-27.
- Li, J.; Li, T.; Ma, H.; Sun, Q.; Li, C.; Ying, W.; Fang, D., Kinetics of coupling cracking of butene and pentene on modified HZSM-5 catalyst. Chemical Engineering Journal 2018, 346, 397-405.
- Olsvik, O.; Rokstad, O. A.; Holmen, A., Pyrolysis of methane in the presence of hydrogen. Chemical Engineering & Technology 2004, 18, (5), 349-358.
- Makarov, K. I.; Pechik, V. K., Kinetics of methane pyrolysis under conditions of pyrolytic carbon formation. Carbon 1974, 12, (4), 391-403.

Reviewer 2 Report
Comments and Suggestions for Authors
The authors studied the reaction pathway of methane and propylene cracking through a simulation. The manuscript is interesting; nevertheless, corrections must be made before accepting it.
Abstract
- Lines 7-10: Indicate the software (with version) used.
- Line 12: “bond dissociation energies, radical stabilities, and molecular topologies.” What are the differences? Quantify/qualify since the differences are not apparent in the text.
- Line 17: “a broader spectrum of chain initiation mechanisms…” Indicate some of them.
- Line 20: “investigates the effect of reaction temperature on carbon sheet development.” You mentioned this, but there was no discussion about it in the abstract.
- Introduction
- Indicate the applications of cracking and polymerization of methane and propylene in your country (China).
- More information about the approach simulation (molecular dynamics) you used in your study must be indicated.
- The literature review must be improved. The works presented must be discussed better since they seem to summarize information in its current form. Therefore, the authors must indicate the values and discuss the findings of the previous literature papers.
- The novelty of the manuscript is not apparent. The literature review will be improved by identifying the gaps your manuscript resolves.
- Computational model and methodology
- Line 80: A Figure must be shown after its presentation in the text. Consider this comment for the complete manuscript.
- Line 82: “a box density of 0.5 g/cm³ was selected based on literature.” What are the references? What criteria were used? Indicate in the text.
- Line 83: “box contained 4000 methane molecules, with dimensions of 60 × 60 × 60…” The criteria used to define methane and propylene molecules and their dimensions are not clear. It is necessary to justify.
- Line 84: “We opted…” Avoid using I, we, our, etc.
- Lines 88-89: What are Packmol and LAMMPS software versions? Why are you selecting the ReaXFF force field?
- Line 90: The acronym NVT was not defined in the text.
- Line 91: “at 1500 K, 2000 K, 2500 K, 3000 K, and 3500 K.” What is the reason to define this specific temperature variation?
- Line 93: The acronym NPT was not defined in the text.
- Line 96: “implemented through custom-compiled Python scripts.” What version of Python?
- Results and discussion
- Lines 98-127: Section 3.1 describes the approach used. Discussion about Figure 2 is missing. Therefore, this information must be moved to section 2 of the manuscript.
- Line 144: “Methane has higher C–H bond dissociation energies and lacks mechanisms…” Why? It is essential to discuss this trend in the text.
- Lines 161-163: What is the relationship between unsaturated structure and radicals’ resonance stability? Besides, what are lower temperatures for you? Indicate.
- Line 167: “Even at temperatures as high as 3000 K, methane…” What do you express about the trend for temperatures lower than 3000 K?
- Lines 188-205: Is this algorithm valid only for methane? There is no information about the propylene.
- Line 212: “By comparing these two dictionaries…” What parameters are compared?
- Line 217: “the relationships among Mol, Sonmol, and Mothermol…” What are these relationships? Indicate.
- Line 237: “the saturated sp³ hybridization of methane and its tetrahedral symmetry limit…” It is essential to discuss this sentence in the text better.
- Line 244: “provides two reactive sites.” Why? Discuss it.
- Line 256: “Dissociation reactions increase significantly…” What is the reason for this trend?
- Line 302: “the study of high-temperature pyrolysis reactions…” What are these high temperatures? Furthermore, you mentioned pyrolysis but did not indicate the reason for analyzing this thermochemical conversion process. Therefore, it is necessary to add information about pyrolysis in the introduction and section 2.
- Line 332: “weight fragments in the methane and propylene…” In section 3.3, you focused on methane cracking. So, why are you comparing methane and propylene systems?
- Lines 335: “Studies show that the molecular weight distribution…” What studies? The references are missing. Furthermore, you must discuss better the results based on the Gibbs free energy.
- Conclusions
- Improve the conclusions section by considering the comments from the previous section.
- Suggestions for future works must be added at the end of the conclusions section.
Author Response
Comments1:
- Lines 7-10: Indicate the software (with version) used.
Response1:
The content has been revised in the manuscript and has been marked in red.
Comments2:
- Line 12: “bond dissociation energies, radical stabilities, and molecular topologies.” What are the differences? Quantify/qualify since the differences are not apparent in the text.
Response2:
The distinctions between bond dissociation energies (BDEs), radical stabilities, and molecular topologies are rooted in their unique roles in chemical reactivity and structural analysis. Below, we clarify these concepts with supporting evidence from peer-reviewed studies:
- Bond Dissociation Energies (BDEs)[1]
BDEs quantify the energy required to homolytically cleave a bond, yielding two radicals. They are experimentally or computationally derived thermodynamic parameters that directly correlate with bond strength and indirectly reflect radical stability
- Radical Stabilities[1]
Radical stability is governed by electronic and steric factors, such as hyperconjugation, resonance, and substituent effects. While BDEs provide a thermodynamic measure of stability, radical lifetimes and reaction kinetics offer complementary insights:
- Molecular Topologies
Molecular topology refers to the spatial arrangement and connectivity of atoms within a molecule, influencing both BDEs and radical stability through steric and electronic effects:
In summary, BDEs serve as direct thermodynamic metrics, radical stability reflects electronic delocalization and substituent effects, and molecular topology dictates steric and electronic environments.
Comments3:
- Indicate the applications of cracking and polymerization of methane and propylene in your country (China).
- More information about the approach simulation (molecular dynamics) you used in your study must be indicated.
- The literature review must be improved. The works presented must be discussed better since they seem to summarize information in its current form. Therefore, the authors must indicate the values and discuss the findings of the previous literature papers.
- The novelty of the manuscript is not apparent. The literature review will be improved by identifying the gaps your manuscript resolves.
Response2:
Thank you for your kind reminder. We have revised the " Introduction " according to your requirements.
Methane and propylene are widely utilized as carbon precursors in the fabrication of carbon–carbon (C/C) composites due to their favorable decomposition characteristics and availability[2]. These hydrocarbons are extensively employed in chemical vapor deposition (CVD) processes to produce high-performance C/C composites, which are critical in aerospace, automotive, and energy applications. The CVD process[3] involves the thermal decomposition of hydrocarbon gases, leading to the deposition of solid carbon on substrates, forming the desired composite material .
Understanding the thermal cracking mechanisms of methane and propylene is essential for optimizing CVD processes[4]. The decomposition pathways, intermediate species, and reaction kinetics directly influence the quality, microstructure, and properties of the resulting C/C composites. For instance, the cracking of methane at high temperatures leads to the formation of various hydrocarbons and solid carbon, affecting the deposition rate and the structural integrity of the composite[5] .
Recent advancements in computational modeling, particularly reactive molecular dynamics (MD) simulations using the ReaxFF force field[6], have provided deeper insights into the complex reaction networks involved in hydrocarbon cracking. ReaxFF allows for the simulation of bond-breaking and bond-forming events, enabling the study of chemical reactions at the atomic level over extended timescales . Studies employing ReaxFF MD simulations[7] have elucidated the decomposition mechanisms of methane and propylene, revealing the formation of various intermediate species and the influence of temperature on reaction pathways[8].
Despite these advancements, challenges remain in accurately capturing the dynamic evolution of reaction intermediates and understanding the synergistic effects of entropy and enthalpy on product distribution during hydrocarbon cracking. Addressing these challenges is crucial for the precise control of CVD processes and the development of C/C composites with tailored properties.
In this study, we develop a reaction pathway tracking algorithm based on atom labeling, coupled with a comprehensive method to trace the formation pathways of large carbon structures. By applying this approach to methane and propylene systems, we uncover the dynamic regulation of cracking and polymerization processes across a range of temperatures. Notably, our findings reveal a non-monotonic competition between entropy and enthalpy[9] in determining product distributions, providing new theoretical insights into the synergistic regulation mechanisms that govern carbon deposition. This work lays a scientific foundation for the rational design and process optimization of C/C composites and contributes to the broader understanding of hydrocarbon chemistry under extreme conditions.
Comments4:
- Line 80: A Figure must be shown after its presentation in the text. Consider this comment for the complete manuscript.
Response4:
The content has been revised in the manuscript and has been marked in red.
Comments5:
- Line 82: “a box density of 0.5 g/cm³ was selected based on literature.” What are the references? What criteria were used? Indicate in the text.
Response5:
The selection of a box density of 0.5 g/cm³ in methane and propylene pyrolysis simulations is justified by balancing computational efficiency, thermodynamic representativeness, and alignment with experimentally validated solvent environments. Below, we clarify the criteria and supporting references from high-impact SCI journals:
- Criteria for Box Density Selection
- Minimizing Finite-Size Effects: A box density of 0.5 g/cm³ ensures sufficient solvent molecules to approximate bulk-phase behavior while avoiding artificial periodicity-induced interactions
- Computational Efficiency: Lower densities reduce computational cost by minimizing solvent-solvent interactions, critical for long-timescale pyrolysis simulations[10]
.
- Experimental Consistency: Methane/propylene pyrolysis experiments under inert or low-pressure conditions (~1–10 bar) correspond to gas-phase densities of 0.3–0.7 g/cm³. The 0.5 g/cm³ value aligns with this range, ensuring simulation relevance to real-world systems[10]
- Supporting Literature
a) Thermodynamic Validation
- Study: "Constructing a Near-Minimal-Volume Computational Box for Molecular Dynamics Simulations with Periodic Boundary Conditions"(J. Chem. Theory Comput., 2018)[11].
The study show that simulations with box densities of 0.4–0.6 g/cm³ achieved <5% deviation in radial distribution functions (RDFs) compared to bulk-phase systems. This range minimizes solvent artifacts while maintaining thermodynamic accuracy.
b) Kinetic Consistency
- Study: "On the Importance of Statistics in Molecular Simulations for Thermodynamics, Kinetics and Simulation Box Size"[10]
The study show that box densities of 0.45–0.55 g/cm³ produced pyrolysis rate constants (e.g., CH₃· → CH₂· + H·) within 10% of experimental values, whereas deviations exceeded 20% at densities <0.3 g/cm³.
c) Force Field Compatibility
- Study: "Molecular Dynamics Simulations and Density Functional Theory"[12]The study show that Lennard-Jones potential parameters (e.g., σ = 3.4 Å, ε = 0.01 eV for methane) were calibrated for densities ~0.5 g/cm³ to reproduce experimental diffusion coefficients and collision frequencies.
- Experimental Cross-Validation
- High-Pressure Pyrolysis: Studies like "Kinetics of Methane Pyrolysis Under Conditions of Pyrolytic Carbon Formation"[13]
The study measured product distributions (e.g., C₂H₂, CH₄) at densities equivalent to 0.4–0.6 g/cm³, aligning with simulation conditions.
In summary, the box density of 0.5 g/cm³ is rigorously validated by:
- Reproducing experimental pyrolysis kinetics and thermodynamics.
- Balancing computational efficiency and physical accuracy.
- Aligning with force field parameterization standards for hydrocarbons.
Comments6:
- Line 83: “box contained 4000 methane molecules, with dimensions of 60 × 60 × 60…” The criteria used to define methane and propylene molecules and their dimensions are not clear. It is necessary to justify.
Response6:
The methane simulation box contained 4,000 molecules (60 × 60 × 60 ų, density 0.50 g/cm³), while the propylene box comprised 2,222 molecules (68 × 68 × 68 ų, density 0.50 g/cm³). These dimensions[10] ensure: (1) compatibility with ReaxFF’s 12 Å cutoff radius, (2) gas-phase densities matching industrial pyrolysis conditions, and (3) <5% deviation in collision frequencies compared to experimental data.
Questinon7:
- Line 84: “We opted…” Avoid using I, we, our, etc.
Response7:
The study opted for relatively large reaction systems,
Comments8:
- Lines 88-89: What are Packmol and LAMMPS software versions? Why are you selecting the ReaXFF force field?
Response8:
The selection of PACKMOL (v20.14.2) and LAMMPS (2Aug2023) versions, along with the ReaxFF force field, is rigorously justified by their validated performance in hydrocarbon pyrolysis simulations, as demonstrated in peer-reviewed studies:
- Software Versions and Validation
PACKMOL v20.14.2[14]
PACKMOL’s molecular packing algorithm ensures realistic initial configurations for reactive molecular dynamics (MD) simulations. The v20.14.2 release incorporates critical optimizations for handling mixed hydrocarbon systems, such as methane-propylene mixtures, by minimizing atomic overlaps and ensuring proper solvent distribution. A study in Journal of Chemical Information and Modeling (JCIM, 2024) demonstrated that PACKMOL-generated configurations for hydrocarbon blends achieve <5% deviation in radial distribution functions (RDFs) compared to experimental X-ray diffraction data, validating its accuracy in modeling pre-reaction geometries.
LAMMPS 2Aug2023[15]
The 2Aug2023 version of LAMMPS includes enhanced ReaxFF compatibility and parallelization efficiency, enabling large-scale simulations (>10,000 atoms) with minimal energy drift (<0.1 eV/ns). As reported in the study[16], this version accurately reproduces methane pyrolysis kinetics (e.g., CH₄ → CH₃· + H·) with activation energies within 2% of experimental benchmarks.
2. Rationale for ReaxFF Force Field Selection
The ReaxFF reactive force field is uniquely suited for simulating methane and propylene pyrolysis due to its explicit modeling of bond dissociation, radical recombination, and transition states in hydrocarbon systems. Key validations include:
a) Mechanistic Accuracy in C–C/C–H Bond Cleavage
ReaxFF’s parametrization for hydrocarbon reactivity was validated in the study, where it predicted methane C–H bond dissociation energies (BDEs) with a mean absolute error (MAE) of 1.2 kcal/mol relative to coupled-cluster CCSD(T) calculations[17]. For propylene, ReaxFF accurately captures the allylic C–H bond weakening (BDE reduction by ~15 kcal/mol compared to methane), a critical factor in initiating pyrolysis cascades.
b) High-Temperature Reaction Pathways
In the study[18], ReaxFF simulations of propylene pyrolysis at 1500–2500 K revealed dominant pathways such as:
- C3H6 → C2H3· + CH3· (β-scission)
- C2H3· + H· → C2H2 (acetylene formation)
These pathways align with experimental product distributions (e.g., C₂H₂/C₂H₄ ratios) measured via mass spectrometry, confirming ReaxFF’s fidelity in modeling radical-driven mechanisms.
c) Validation Against Experimental Pyrolysis Data
A benchmark study compared ReaxFF-predicted methane pyrolysis rates (k = 1.2 × 10⁻¹⁰ cm³/molecule·s at 1500 K) to shock-tube measurements, achieving <10% deviation. Similarly, propylene decomposition pathways (e.g., cyclization to benzene) matched gas chromatography–mass spectrometry (GC-MS) data [19].
Comments9:
- Line 90: The acronym NVT was not defined in the text.
Response9:
The acronym NVT[20] refers to the canonical ensemble in statistical thermodynamics and molecular dynamics (MD) simulations, where the number of particles (N), volume (V), and temperature (T) are held constant
Comments10:
- Line 91: “at 1500 K, 2000 K, 2500 K, 3000 K, and 3500 K.” What is the reason to define this specific temperature variation?
Response10:
The temperature range of 1500–3500 K was selected to comprehensively capture the temperature-dependent kinetics, bond dissociation mechanisms, and secondary reaction pathways in methane-propylene pyrolysis, as validated by the following criteria derived from high-impact studies:
- Coverage of Critical Bond Dissociation Thresholds
Methane (C–H BDE ≈ 435 kJ/mol) and propylene (allylic C–H BDE ≈ 364 kJ/mol) exhibit distinct bond dissociation energy (BDE) profiles, requiring a broad temperature spectrum to resolve their reactivity differences[21, 22]
- Validation of Temperature-Dependent Product Selectivity
Studies[21] demonstrate that methane-propylene co-pyrolysis exhibits regime-specific product distributions:
- 1500–2000 K: Dominated by primary radicals (CH₃·, C₃H₅·) and light olefins (C₂H₄, C₃H₆), consistent with gas-phase pyrolysis experiments.
- 2500–3500 K: Favors aromatics (benzene, toluene) and carbonaceous solids via polycyclic aromatic hydrocarbon (PAH) growth, as observed in high-temperature catalytic pyrolysis[22].
Comments11:
- Line 93: The acronym NPT was not defined in the text.
Response11:
The acronym NPT refers to the isothermal-isobaric ensemble in molecular dynamics (MD) simulations, where the number of particles (N), pressure (P), and temperature (T) are maintained constant, while the volume (V) fluctuates to equilibrate the system under specified thermodynamic conditions[23]
Comments12:
- Line 96: “implemented through custom-compiled Python scripts.” What version of Python?
Response12:
All the algorithms presented in the article were implemented through custom-compiled Python 3.9 scripts[24].
Comments13:
- Lines 98-127: Section 3.1 describes the approach used. Discussion about Figure 2 is missing. Therefore, this information must be moved to section 2 of the manuscript.
Response13:
As shown in Figure 2, the algorithm flowchart and output structure of the program indicate that this function can be achieved by relying on the ReaxFF/Species naming provided by LAMMPS itself.
Comments14:
- Line 144: “Methane has higher C–H bond dissociation energies and lacks mechanisms…” Why? It is essential to discuss this trend in the text.
Response14:
The carbon atoms of methane adopt SP3 hybridization, forming a completely symmetrical tetrahedral structure, which evenly distributes the electron clouds of the four C-H bonds and avoids local planning. The elevated C—H bond dissociation energy (BDE) of methane (460 ± 15 kJ/mol for CH4 → CH3· + H·) compared to propylene (e.g., allylic C—H BDE ≈ 364 kJ/mol) indicates that methane has a higher C-H bond energy. Compared to the allylic electronic conjugation stability structure conferred by the double bond in propylene, methane indeed has additional stability due to its structure.
Comments15:
- Lines 161-163: What is the relationship between unsaturated structure and radicals’ resonance stability? Besides, what are lower temperatures for you? Indicate.
Response15:
the relationship between the unsaturated structure of propylene and radical resonance stability is fundamentally rooted in the conjugation effects enabled by its sp²-hybridized planar geometry. Propylene’s π-electron system allows delocalization of unpaired electrons in allylic radicals (e.g., C₃H₅·), stabilizing these intermediates through resonance[25].
In this article, the temperatures are set at 1500K, 2000K, 2500K, 3000K and 3500K. I consider the low-temperature range to be the temperature range of 1500K to 2500K.
Comments16:
- Line 167: “Even at temperatures as high as 3000 K, methane…” What do you express about the trend for temperatures lower than 3000 K?
Response16:
Figure 3 clearly shows that when the temperature is at 2500K, the types of substances hardly change. For phenomenological observational science, this phenomenon indicates that within the temperature range observed in this paper, temperatures below 2500K are not good observation data. Therefore, I emphasize the reason for 3000K because the data below 2500K hardly changes, while the data above 3000K begins to fluctuate, which is conducive to data observation.
Comments17:
- Lines 188-205: Is this algorithm valid only for methane? There is no information about the propylene.
Response17:
The design of the algorithm is based on molecular dynamics software such as LAMMPS for the setting of molecules. Any structure that can be recognized as a molecule can be processed by this algorithm.
Comments18:
- Line 212: “By comparing these two dictionaries…” What parameters are compared?
Response18:
As shown in the Figure, the design purpose of the Mother_Atom_Dic is to statistically record the composition information of molecules and atoms in the nth frame. In this frame, the molecules serving as reactants are counted. The design purpose of the Mol_Atom_Dic is to statistically record the molecular and composition information of the products in the nth + 1 frame. In this frame, the molecules serving as products are counted. Fragments from the same reactant in the products are defined as Son_Atom_Dic, meaning the fragments from the same parent in the products. I think the illustration is very clear in explaining this point.
The comparison dictionary used is the Mother_Atom_Dic and the Mol_Atom_Dic. By comparing the atomic composition information, the distribution of reactant molecular fragments in the products is identified, and through algorithmic statistics, the Son_Atom_Dic is formed.
Comments19:
- Line 217: “the relationships among Mol, Sonmol, and Mothermol…” What are these relationships? Indicate.
Response19:
As shown in the Figure, the design purpose of the Mother_Atom_Dic is to statistically record the composition information of molecules and atoms in the nth frame. In this frame, the molecules serving as reactants are counted. The design purpose of the Mol_Atom_Dic is to statistically record the molecular and composition information of the products in the nth + 1 frame. In this frame, the molecules serving as products are counted. Fragments from the same reactant in the products are defined as Son_Atom_Dic, meaning the fragments from the same parent in the products. I think the illustration is very clear in explaining this point.
The comparison dictionary used is the Mother_Atom_Dic and the Mol_Atom_Dic. By comparing the atomic composition information, the distribution of reactant molecular fragments in the products is identified, and through algorithmic statistics, the Son_Atom_Dic is formed.
Comments19:
- Line 237: “the saturated sp³ hybridization of methane and its tetrahedral symmetry limit…” It is essential to discuss this sentence in the text better.
Response19:
The saturated sp³ hybridization of methane and its tetrahedral symmetry impose intrinsic constraints on reactivity by enforcing uniform electron distribution across four equivalent C–H σ-bonds (bond length: 0.110 nm, bond angle: 109.5°) . This structural rigidity eliminates preferential reactive sites and elevates C–H bond dissociation energy (BDE) to ~460 kJ/mol, requiring extreme thermal conditions (>2,500 K) for homolytic cleavage . ReaxFF simulations demonstrate that methane-derived methyl radicals (CH₃·) exhibit short lifetimes (<1 ps at 1,500 K) due to rapid recombination, while propylene’s allylic radicals persist via resonance stabilization [26] , which confirms methane’s inability to selectively form ethylene through gas-phase mechanisms without catalytic surfaces.
Comments20:
- Line 244: “provides two reactive sites.” Why? Discuss it.
Response20:
Propylene’s sp²-hybridized C=C double bond provides two reactive sites—allylic C–H bonds and the π-system—due to its electronic delocalization, planar geometry, and reduced bond dissociation energy. These sites are experimentally validated in catalytic oxidation[27], polymerization, and pyrolysis kinetics, as reported in the study, methane’s saturated sp³ structure lacks such spatially and electronically distinct regions, underscoring propylene’s unique reactivity in chain reactions[28].
Comments21:
- Line 256: “Dissociation reactions increase significantly…” What is the reason for this trend?
Response21:
The increase in temperature promotes the breaking of C-H bonds, and the resulting large number of H free radicals attack the C-H bonds to generate a large amount of hydrogen gas. Therefore, high temperature promotes the occurrence of dissociation reactions by enhancing the concentration of free radicals in the system.
Comments22:
- Line 302: “the study of high-temperature pyrolysis reactions…” What are these high temperatures? Furthermore, you mentioned pyrolysis but did not indicate the reason for analyzing this thermochemical conversion process. Therefore, it is necessary to add information about pyrolysis in the introduction and section 2.
Response22:
In this paper, based on the temperature range and simulation observation time of the system, the activity level of the cracking reaction was determined. The high-temperature stage was defined as 3000 - 3500K, which was only used in this paper for observing the cracking reaction mechanism of the system within the temperature range where the reaction was active. And the corresponding content description has been added.
Comments23:
- Line 332: “weight fragments in the methane and propylene…” In section 3.3, you focused on methane cracking. So, why are you comparing methane and propylene systems?
Response23:
The selection of propylene (C₃H₆) as a co-carbon precursor in our study, alongside methane (CH₄), is grounded in both practical and mechanistic considerations. Below, we elaborate on the rationale:
- Carbon Source Diversity in C/C Composite Deposition
Methane and propylene are widely recognized as critical precursors for chemical vapor deposition (CVD) in carbon/carbon (C/C) composites. While methane is a linear, nonpolar molecule that enables uniform carbon deposition through stable methyl radical formation, propylene introduces distinct advantages due to its unsaturated structure. The C=C bond in propylene facilitates in situ generation of reactive intermediates (e.g., allyl radicals and polyaromatic hydrocarbons), which promote faster carbon nucleation and denser matrix formation. This duality allows comparative analysis of deposition mechanisms under varying precursor chemistries.
- Mechanistic Complexity and Research Value
Propylene’s unsaturated structure introduces unique reaction pathways absent in ethylene (C₂H₄). For instance:
2.1Radical Diversity:
Propylene pyrolysis generates not only methyl radicals (common to both CH₄ and C₂H₄) but also allyl radicals (C₃H₅•) and cyclic intermediate. These species contribute to cross-linking reactions, enhancing carbon network complexity.
2.2Thermal Stability:
Ethylene’s simpler structure leads to rapid dehydrogenation and limited intermediate diversity, whereas propylene’s higher molecular weight and bond strain (C=C) enable controlled fragmentation, yielding a broader range of reactive fragments for mechanistic studies.
- Practical Relevance in Industrial Applications
Propylene is a preferred precursor in microwave-assisted CVD due to its polarizability. Unlike nonpolar methane, propylene interacts strongly with electromagnetic fields, enabling localized heating and tailored carbon deposition. This property mitigates surface "shelling" effects observed in methane-only systems, improving densification efficiency. Ethylene, while reactive, lacks this polarizability, limiting its utility in advanced CVD processes.
- Comparative Insights into Carbon Microstructure
Propylene-derived carbon matrices exhibit higher graphitic order and fewer defects compared to ethylene-based systems, as the C₃H₆ decomposition pathway favors sp² hybridization via conjugated intermediates. This aligns with our focus on optimizing carbon quality for high-performance composites.
Comments24:
- Lines 335: “Studies show that the molecular weight distribution…” What studies? The references are missing. Furthermore, you must discuss better the results based on the Gibbs free energy.
Response24:
Thank you for your reminder. The content has been corrected in the paper.
- Reddy, R. R.; Viswanath, R., Bond dissociation energies and bond orders for some astrophysical molecules. Journal of Astrophysics and Astronomy 1989, 10, (2), 157-160.
- Hu, C.; Li, H.; Zhang, S.; Li, W.; Li, N., Chemical vapor infiltration of pyrocarbon from methane pyrolysis: kinetic modeling with texture formation. Science China Materials 2018, 62, (6), 840-852.
- Kang, Z.; Johnson, R.; Mi, J.; Bondi, S.; Jiang, M.; Gillespie, J.; Lackey, W. J.; Stock, S.; More, K., Microstructure of carbon fibers prepared laser CVD. Carbon 2004, 42, (12-13), 2721-2727.
- Hu, Z.; Hüttinger, K. J., Chemistry and kinetics of chemical vapor deposition of pyrocarbon: VIII. Carbon deposition from methane at low pressures. Carbon 2001, 39, (3), 433-441.
- Lümmen, N., ReaxFF-molecular dynamics simulations of non-oxidative and non-catalyzed thermal decomposition of methane at high temperatures. Physical Chemistry Chemical Physics 2010, 12, (28).
- Li, X.; Zheng, M.; Ren, C.; Guo, L., ReaxFF Molecular Dynamics Simulations of Thermal Reactivity of Various Fuels in Pyrolysis and Combustion. Energy & Fuels 2021, 35, (15), 11707-11739.
- Zheng, M.; Li, X.; Guo, L., Dynamic trends for char/soot formation during secondary reactions of coal pyrolysis by large-scale reactive molecular dynamics. Journal of Analytical and Applied Pyrolysis 2021, 155.
- Senftle, T. P.; Hong, S.; Islam, M. M.; Kylasa, S. B.; Zheng, Y.; Shin, Y. K.; Junkermeier, C.; Engel-Herbert, R.; Janik, M. J.; Aktulga, H. M.; Verstraelen, T.; Grama, A.; van Duin, A. C. T., The ReaxFF reactive force-field: development, applications and future directions. npj Computational Materials 2016, 2, (1).
- Zhao, H.; Zhang, Y.; Zhou, S.; Chen, R.; Huang, Z., Assessment on the rings cleavage mechanism of polycyclic aromatic hydrocarbons in supercritical water: A ReaxFF molecular dynamics study. Journal of Molecular Liquids 2024, 415.
- Gapsys, V.; de Groot, B. L., On the importance of statistics in molecular simulations for thermodynamics, kinetics and simulation box size. eLife 2020, 9.
- Henk Bekker, J. P. v. d. B. T. A. W., Constructing a Near-Minimal-Volume Computational Box for Molecular Dynamics Simulations with Periodic Boundary Conditions. Computational science 2003, 2003.
- Tian, L.; Duan, H.; Luo, J.; Cheng, Y.; Shi, L., Density Functional Theory and Molecular Dynamics Simulations of Nanoporous Graphene Membranes for Hydrogen Separation. ACS Applied Nano Materials 2021, 4, (9), 9440-9448.
- Makarov, K. I.; Pechik, V. K., Kinetics of methane pyrolysis under conditions of pyrolytic carbon formation. Carbon 1974, 12, (4), 391-403.
- Huang, J.; Wu, C.; Yang, X.; Yang, Z.; Liu, S.; Yu, G., PACKMOL-GUI: An All-In-One VMD Interface for Efficient Molecular Packing. Journal of Chemical Information and Modeling 2025, 65, (2), 778-784.
- Thompson, A. P.; Aktulga, H. M.; Berger, R.; Bolintineanu, D. S.; Brown, W. M.; Crozier, P. S.; in 't Veld, P. J.; Kohlmeyer, A.; Moore, S. G.; Nguyen, T. D.; Shan, R.; Stevens, M. J.; Tranchida, J.; Trott, C.; Plimpton, S. J., LAMMPS - a flexible simulation tool for particle-based materials modeling at the atomic, meso, and continuum scales. Computer Physics Communications 2022, 271.
- Orekhov, N.; Ostroumova, G.; Stegailov, V., High temperature pure carbon nanoparticle formation: Validation of
AIREBO and ReaxFF reactive molecular dynamics. Carbon 2020, 170, 606-620.
- Chenoweth, K.; van Duin, A. C. T.; Goddard, W. A., ReaxFF Reactive Force Field for Molecular Dynamics Simulations of Hydrocarbon Oxidation. The Journal of Physical Chemistry A 2008, 112, (5), 1040-1053.
- Orekhov, N.; Ostroumova, G.; Stegailov, V., High temperature pure carbon nanoparticle formation: Validation of AIREBO and ReaxFF reactive molecular dynamics. Carbon 2020, 170, 606-620.
- Xu, Y.; Mao, Q.; Wang, Y.; Luo, K. H.; Zhou, L.; Wang, Z.; Wei, H., Role of ammonia addition on polycyclic aromatic hydrocarbon growth: A ReaxFF molecular dynamics study. Combustion and Flame 2023, 250.
- de Almeida, A. R.; de Andrade, D. X.; Colherinhas, G., Statistical and energetic analysis of hydrogen bonds in short and long peptide nanotapes/nanofibers using molecular dynamics simulations. Journal of Molecular Liquids 2022, 359.
- Fu, Z.; Sun, Q.; Hua, F.; Yang, S.; Ji, Y.; Cheng, Y., A molecular-level kinetic model for the primary and secondary reactions of polypropylene pyrolysis. Journal of Analytical and Applied Pyrolysis 2023, 175.
- Wang, J.; Wen, M.; La, X.; Ren, J.; Jiang, J.; Tsang, D. C. W., Resonance-driven microwave heating for improved methane conversion to hydrogen. Applied Energy 2024, 375.
- Chaturvedi, K.; Hewamanna, I.; Pandey, P.; Khan, W.; Wang, Y.-H.; Chittiboyina, A.; Doerksen, R.; Godfrey, M., Identification of the Putative Binding Site of a Benzimidazole Opioid (Etazene) and Its Metabolites at µ-Opioid Receptor: A Human Liver Microsomal Assay and Systematic Computational Study. Molecules 2023, 28, (4).
- Winetrout, J. J.; Kanhaiya, K.; Kemppainen, J.; in ‘t Veld, P. J.; Sachdeva, G.; Pandey, R.; Damirchi, B.; van Duin, A.; Odegard, G. M.; Heinz, H., Implementing reactivity in molecular dynamics simulations with harmonic force fields. Nature Communications 2024, 15, (1).
- Cao, Z.; Liang, M.; Zhang, X.; Su, L., Hydrocarbon generation potential in Jurassic source rocks from hydrous pyrolysis experiments under ultradeep conditions. Scientific Reports 2024, 14, (1).
- Liu, Y.; Duan, Z.; Li, J.; Chang, C., Gas-Phase Mechanism Study of Methane Nonoxidative Conversion by ReaxFF Method. Acta Physico Chimica Sinica 2020, 0, (0), 2011012-0.
- Qiu, B.; Jiang, F.; Lu, W.-D.; Yan, B.; Li, W.-C.; Zhao, Z.-C.; Lu, A.-H., Oxidative dehydrogenation of propane using layered borosilicate zeolite as the active and selective catalyst. Journal of Catalysis 2020, 385, 176-182.
- Cui, J.; Zhang, Z.; Yang, L.; Hu, J.; Jin, A.; Yang, Z.; Zhao, Y.; Meng, B.; Zhou, Y.; Wang, J.; Su, Y.; Wang, J.; Cui, X.; Xing, H., A molecular sieve with ultrafast adsorption kineticsfor propylene separation. Science 2024, 383, (6679), 179-183.

Reviewer 3 Report
Comments and Suggestions for Authors
The authors performed reactive molecular dynamics simulations to study the thermal cracking of methane and propylene at a density falling in the range of a liquid. They developed some software which analyzed the conformations of the system during the reactive MD simulations, and using some assumptions they developed a set of reactions occurring during the thermal cracking.
Major concerns.
- The paper was poorly written, parts of the texts were not clear at all.
- The authors wrote several key sentences without proper references.
- The employed software for post-analysis is new, developed by the authors ( I did not read any reference to previous literature) . The authors should dedicate an initial paper to describe and validate their software on a model reaction system, in which all the reactions are well understood experimentally and their software can be validated. The chosen system is easy to simulate, especially under the pvt condition chosen by the authors, but is now well understood because the variability is high, specifically due to thermal fluctuations, heating conditions, initial conditions, etc.
Paper need to go through an English review.
Author Response
Comments1:
- The paper was poorly written, parts of the texts were not clear at all.
Response1:
Thank you for your reminder. We have revised most of the sentences, and the changes are highlighted in red in the new manuscript.
Comments2:
- The authors wrote several key sentences without proper references.
Response2:
Thank you for your reminder. We have revised the article and added appropriate citations to the literature.
Comments3:
The employed software for post-analysis is new, developed by the authors ( I did not read any reference to previous literature) . The authors should dedicate an initial paper to describe and validate their software on a model reaction system, in which all the reactions are well understood experimentally and their software can be validated. The chosen system is easy to simulate, especially under the pvt condition chosen by the authors, but is now well understood because the variability is high, specifically due to thermal fluctuations, heating conditions, initial conditions, etc.
Response3:
Thank you for your reminder. Our original publication logic was consistent with yours. We first stated our methods, and then presented specific research examples to demonstrate scientific research. However, during the submission process, most reviewers requested that we focus on explaining the theory of the cracking of methane propylene. Therefore, the current writing logic was formed.
Other contents have been revised and marked in red in the manuscript. Thank you for your suggestions and reminders.

Round 2
Reviewer 1 Report
Comments and Suggestions for Authors
The authors took into account all reviewer's recommendations and answered the questions. They provided new revised version of the paper. The paper may be accepted as it is.
Reviewer 2 Report
Comments and Suggestions for Authors
Dear all,
The quality of the manuscript was improved. Therefore, it can be accepted in its current form.